# rSO2 Measurement Using NIRS for Lower-Limb Blood Flow Monitoring and Estimation of Safe Balloon Occlusion/Deflation Time in Patients with PAS Who Underwent PBOA during CS

**DOI:** 10.3390/medicina59061146

**Published:** 2023-06-14

**Authors:** Hiroyuki Tokue, Azusa Tokue, Yoshito Tsushima

**Affiliations:** Department of Diagnostic and Interventional Radiology, Gunma University Hospital, 3-39-22 Showa-machi, Maebashi 371-8511, Gunma, Japan

**Keywords:** regional oxygen saturation, near-infrared spectroscosspy, placenta accreta spectrum, prophylactic balloon occlusion of the abdominal artery, lower-limb blood flow

## Abstract

We examined the utility of regional oxygen saturation (rSO2) measurement using near-infrared spectroscopy (NIRS) for monitoring lower-limb blood flow and estimate the safe balloon occlusion/deflation time in patients with PAS who underwent prophylactic balloon occlusion of the abdominal artery (PBOA) during cesarean section (CS). During CS, the NIRS probes were positioned on either of the anterior tibial muscles. rSO2 was measured continuously during balloon occlusion/deflation. A cycle consisted of inflating the aortic balloon for 30 min and deflating it for 5 min. The rSO2 before/during balloon occlusion and after 5 min of balloon deflation were evaluated. Sixty-two lower limbs (fifteen women and data from 31 sessions of balloon inflation/deflation) were evaluated. rSO2 during balloon occlusion was significantly lower than rSO2 before balloon occlusion (57.9% ± 9.6% vs. 80.3% ± 6.0%; *p* < 0.01). There were no significant differences between rSO2 before balloon occlusion and rSO2 after 5 min of balloon deflation (80.3% ± 6.0% vs. 78.7% ± 6.6%; *p* = 0.07). Postoperatively, the lower limbs showed no ischemic symptoms. NIRS can assess lower-limb rSO2 during PBOA for PAS in real time to determine ischemia severity, duration, and recovery capacity.

## 1. Introduction

Placenta accreta spectrum (PAS) is the general term for abnormal placental trophoblast adhesion to the uterine myometrium and includes placenta accreta, increta, and percreta depending on the depth of villous invasion [1]. The incidence of PAS disorders has increased worldwide rapidly in recent years with the increasing number of cesarean sections (CSs) [1]. Traditionally, PAS has been treated with peripartum hysterectomy. In recent years, however, a uterus-preserving strategy has been gradually adopted by physicians and patients who wish to maintain fertility. Prophylactic balloon occlusion of the abdominal artery (PBOA) is one of the effective methods for managing severe hemorrhage caused by PAS [2]. Although there is insufficient evidence, PBOA is safe and effective and reduces intraoperative blood loss and the need for perioperative hemostatic measures, intraoperative red cell transfusions, and hysterectomies [3].

Alternatively, some studies have reported PBOA complications for PAS ranging from 0.9% to 4.4% [4,5]. Lower-limb ischemia, hyperkalemia, and acute kidney injury associated with ischemic reperfusion injury are rare but serious complications of PBOA treatment for PAS. In PBOA, the occlusion time to prevent lower-limb ischemia has been reported to be 20–45 min [6]. Nonetheless, balloons are often used based on the physician’s experience, and adequate balloon occlusion/inflation time and inflation timing have not been investigated objectively using reliable clinical indices.

Recently, noninvasive near-infrared spectroscopy (NIRS) has been used to assess tissue ischemia by measuring regional oxygen saturation (rSO2) [7]. This technique was initially applied clinically to monitor cerebral oxygenation in neonates; today it plays an important role in the diagnosis of hypoxic brain injury and the study of cerebral hemodynamics in preterm infants and neonates [8,9]. Other applications are largely experimental and relate to blood flow and perfusion studies in skeletal muscle and a variety of organs [10]. 

The study aimed to examine the utility of rSO2 measurement using NIRS for monitoring lower-limb blood flow and estimate the safe balloon occlusion/deflation time in patients with PAS who underwent PBOA during CS.

To the best of our knowledge, no study has monitored rSO2 during CS using PBOA for PAS. Using rSO2 monitoring, we hypothesized that we would be able to diagnose lower limb ischemia during vascular occlusion sooner and determine a safe occlusion time. Thus, we monitored lower limb circulation using NIRS during CS with PBOA for PAS.

## 2. Materials and Methods

### 2.1. Patients

The institutional Review Board of Gunma University Hospital approved the study (HS2018-235), and patients’ informed consent was waived on account of the retrospective nature of the study. We retrospectively analyzed the records of pregnant women with PAS who were monitored using NIRS while undergoing PBOA during CS in our institution between January 2018 and April 2023. All cesarean deliveries were planned, and based on the surgical findings, we decided to perform hysterectomy and plan placental removal and uterus reconstruction. In all patients, PBOA was used to preserve the uterus as much as possible.

The inclusion criteria were (1) being diagnosed as PAS based on ultrasound (US) or magnetic resonance imaging and confirmed based on intraoperative findings, (2) the absence of hemorrhage before surgery, (3) a period of gestation > 28 weeks, (4) availability of patient history of previous cesarean delivery, and (5) a preoperative hemoglobin level > 10.0 g/L. The exclusion criteria were (1) severe obstetric complications, especially hematological diseases or coagulation disorders, gestational hypertension, or cardiopulmonary insufficiency, (2) fetal anomaly, fetal growth restriction, or fetal distress, and (3) planned hysterectomy.

### 2.2. Interventional Procedure

PBOA was performed in the digital subtraction angiography operating room, on the day of CS. After administering local anesthesia, the right femoral artery was accessed using Seldinger’s technique, and a 7-French introducer sheath was inserted for placing a 7-French aortic occlusion balloon catheter (Rescue BalloonR, 12–14 mm diameter, Tokai Medical Product, Kasugai, Japan). The balloon was inflated with saline. Once the catheter was placed in the correct position, it was securely taped to the skin. After securing the aortic balloon, the patient was transferred to the operating room for cesarean delivery under spinal anesthesia. Using a mobile C-arm X-ray machine, the correct positioning of the catheter was confirmed just before CS. If necessary, fluoroscopy was performed during CS. As requested by obstetricians, after delivering the infant and clamping the umbilical cord, the occlusion balloon was inflated. Thereafter, the obstetrician surgically excised as much of the placenta as possible, along with any myometrium, and reconstructed the uterus under general anesthesia with tracheal intubation. If hemorrhage from the uterus persisted, the blockage was repeated until the bleeding stopped. The placenta was manually removed as far as possible. If it was difficult to manually remove the placenta, it was left in situ, and a hysterectomy was performed immediately. Furthermore, hysterectomy was also performed when it was difficult to control bleeding in patients during CS. The patients were then observed for 60 min in the recovery room. The balloon catheter was withdrawn after completing the entire procedure. The arterial sheath was removed 6 h after completing the procedure. Manual external compression at the arterial puncture site was performed by interventional physicians to ensure hemostasis.

### 2.3. NIRS Data Acquisition and PBOA Management

The rSO2 of both the lower limbs was percutaneously measured using a continuous dual-infrared spectrometer, INVOS 5100C (Somanetics, Troy, MI, USA). NIRS probes were positioned on either of the anterior tibial muscles during CS. The rSO2 was measured continuously during balloon occlusion/deflation (Figure 1 and Figure 2).

Placement of NIRS optodes at both lower limbs of the patients was performed in the operating theater after anesthesia induction. All patients were measured while in lithotomy position. NIRS measurements of the lower limb were monitored continuously and stored. The moment of balloon occlusion/deflation was noted, as well as the end of operation. Vital parameters, such as blood pressure and heart frequency, were recorded online. Measurements stopped when the patient left the operating theater.

As a rule, the aortic balloon was alternately inflated for 30 min and then deflated for 5 min, which was defined as one cycle. Inflation time was extended depending on the degree of bleeding in the surgical field. When rSO2 measured after 5 min of balloon deflation was less than 90% of rSO2 before balloon occlusion, the deflation time was extended for 5 min.

### 2.4. Data Analysis

We retrospectively examined clinical data from medical records, including age, parity, number of previous CS, degree of placental adhesion, operation time, estimated blood loss (EBL), hysterectomy, amount of packed RBC transfusions, postoperative hospital stay, serum creatinine, serum potassium (K), serum lactate levels, and surgical complications. The operation time for the CS was defined as the time from the initial incision to the completion of wound closure, including a hysterectomy if performed. The EBL was measured based on the volume of the suction canisters in the operating room and the weight of the surgical pads. The evaluation of rSO2 was performed before and during balloon occlusion and after 5 min of balloon deflation, along with the estimation of the balloon occlusion time.

### 2.5. Statistical Analysis

Continuous variables with a normal or non-normal distribution were expressed as the mean ± standard deviation, and categorical variables were expressed as numbers (proportions, %). Student’s *t*-test and the Mann–Whitney U-test were used. We used the SPSS v. 22 software (IBM Corp., Tokyo, Japan) for statistical analysis. Statistical significance was set at *p* < 0.05.

## 3. Results

Sixty-two lower limbs (fifteen women and data from 31 sessions of balloon inflation/deflation; cycles of balloon inflation/deflation were 2.1 ± 0.9 per patient) were evaluated. Continuous rSO2 measurement was possible in all patients. Patient backgrounds and maternal outcomes are shown in Table 1.

There were no PBOA-related complications until discharge, and no damage to the adjacent pelvic organs was observed during the operation. In 8/31 sessions, bleeding and hypotension were observed when the balloon was deflated, but they were treated with blood transfusion. Two out of fifteen patients suffered postoperative complications. In two patients, a hematoma developed near the postoperative site with associated infections, but the condition was resolved after the administration of antibiotics. All mothers and babies were healthy at the time of discharge. In addition, within a year, all the women had resumed their regular menstrual cycles.

The rSO2 and serum laboratory data are shown in Figure 3 and Table 2.

The rSO2 during balloon occlusion was significantly lower than the rSO2 before balloon occlusion (57.9% ± 9.6% vs. 80.3% ± 6.0%; *p* < 0.01). The rSO2 increased in all patients after balloon deflation. There were no significant differences between rSO2 before balloon occlusion and rSO2 after 5 min of balloon deflation (80.3% ± 6.0% vs. 78.7% ± 6.6%; *p* = 0.07). The balloon occlusion time was 30.7 ± 1.6 min. In 5/62 cases (8.1%), the deflation time was extended for 5 min because rSO2 after 5 min of balloon deflation was less than 90% of rSO2 before balloon occlusion. The rSO2 recovered with a 5 min extension of balloon deflation in all patients. Serum laboratory data related to the ischemic change of the lower limbs showed no significant differences when the preoperative values were compared with those on postoperative day one. No discoloration, ischemic symptoms, or neuropathy in the lower limbs were observed postoperatively. No findings suggestive of arterial thrombosis were detected by US or laboratory data.

## 4. Discussion

In the present study, continuous perioperative measurement of rSO2 was possible in all patients without difficulty. NIRS measurement of rSO2 in the lower limbs can assess lower-limb perfusion in real time and may be useful for determining the optimal vascular occlusion/deflation time in PBOA for PAS.

During vascular occlusion by a balloon, it is difficult to detect the pulse of the dorsalis pedis artery with a pulse oximeter or palpation because the pulse wave of the arterial blood flow cannot be detected. NIRS allows continuous noninvasive monitoring of concentration changes occurring in oxy and deoxyhemoglobin (HbO2 and Hb) in the body [11]. Information is provided on changes in tissue oxygen supply and intracellular oxygen availability and utilization. The oxygenation level of the local tissue is determined by NIRS by estimating its perfusion by the entire vascular tree (primarily small vessels and arterioles, capillaries, and venules) [11]. Recently, some studies have evaluated lower extremity perfusion by NIRS, and its usefulness has been demonstrated [7,11,12,13]. In the field of obstetrics and gynecology, NIRS may be useful for monitoring lower extremity perfusion when postpartum hemorrhage is managed with PBOA. The determination of a critical rSO2 cutoff value or a critical period of ischemia is necessary. However, there are limited studies regarding the evaluation of lower extremity blood flow by NIRS in patients with PAS treated with PBOA, and the optimal balloon occlusion/deflation time is unknown.

A previous study proposed the criteria for diagnosis of lower-limb ischemia as rSO2 values in the lower limb < 40% or decreased by 25% or more during venoarterial extracorporeal membrane oxygenation (VA-ECMO). In our study, there were no patients in whom the rSO2 values in the lower limb were <40%, and in 21/50 (42%) cases the rSO2 decreased by 25% or more [12]. However, there was no complication related to ischemia. In the previous study, monitoring of rSO2 by NIRS during aortic aneurysm surgery showed that rSO2 before aortic clamping was 67, and after clamping the aorta or iliofemoral arteries (46 min for clamp time), the rSO2 dropped to 32. After declamping, the rSO2 increased to 74 [13], which was longer than our aortic occlusion time and lower minimum rSO2, but no complications were associated with lower limbs ischemia. Specifically in PAS, we hypothesize that collateral circulation pathways of the inferior renal artery were enabled during PBOA. Therefore, the balloon occlusion time of longer than the 30 min, that was performed in our study, may be tolerated in lower-limb ischemia. Bleeding and hypotension were observed when the balloon was deflated; however, accurate prediction of unacceptable rSO2 and time to deflate may lead to longer vascular occlusion and avoidance of massive bleeding. If the time until rSO2 decrease can be accurately determined, it may be possible to perform longer vascular occlusions and avoid massive bleeding. Further studies with a larger number of patients are required to elucidate these aspects.

This study has some limitations. First, none of our patients had persistent perioperative embolism, lower-limb ischemic changes, or abnormalities in the coagulative/fibrinolytic parameters, and we were unable to prove the added value of NIRS. Additionally, it is not known whether continuous perioperative measurement of rSO2 is genuinely conductive for reducing the incidence of lower-limb ischemia.

Second, this was a single-center retrospective study with a small sample size. This study did not include any evaluations beyond the established inflation/deflation periods. Third, during the use of NIRS, certain parameters (such as anemia, regional blood flow, and vasoconstriction) can influence rSO2 measurement. In fact, as much as possible we tried to preserve the uterus, which resulted in more bleeding than reported in the past [3]. The rSO2 values were obtained consecutively from the same patient. So, there may have been selective bias. Furthermore, this study lacked a long-term follow-up because of its retrospective design. In the future, larger investigations involving multiple centers and large numbers of patients should be performed with longer follow-up periods to provide an accurate assessment and validation of the clinical efficacy of NIRS.

## 5. Conclusions

In conclusion, NIRS can evaluate rSO2 of lower limbs during PBOA for PAS in real time and allows interpretation of the severity and duration of lower-limb ischemia and the recovery capacity from temporary ischemic insults. A 30 min inflation time and a 5 min deflation time were considered acceptable in PAS patients who underwent PBOA during CS. Further evaluation using NIRS in a larger number of cases is needed to determine the optimal balloon occlusion/deflation time.

## Figures and Tables

**Figure 1 medicina-59-01146-f001:**
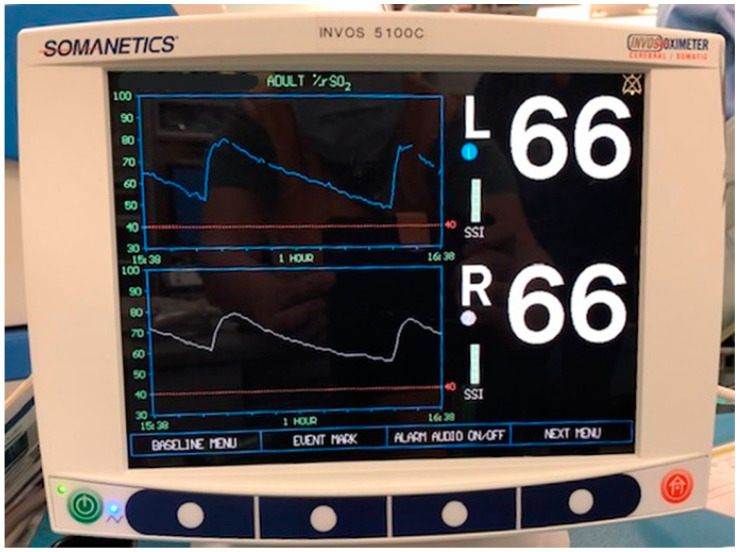
rSO2 was measured continuously during balloon occlusion/deflation using a continuous dual-infrared spectrometer, INVOS 5100C (Somanetics, Troy, MI, USA).

**Figure 2 medicina-59-01146-f002:**
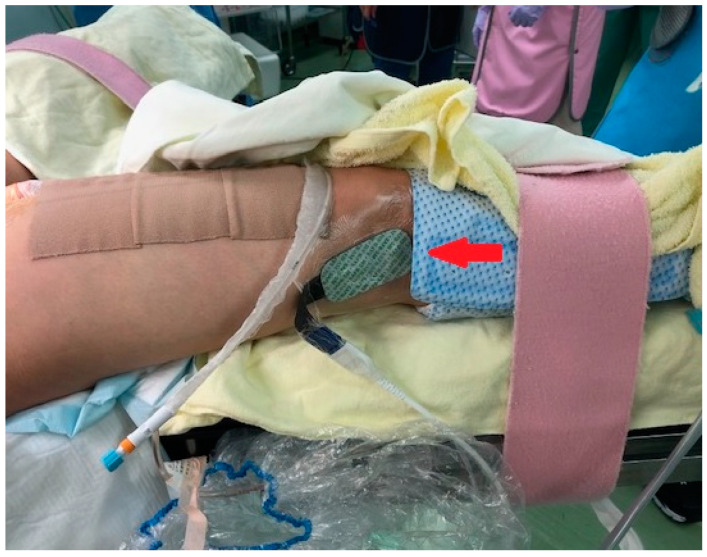
NIRS probes were positioned on either of the anterior tibial muscles during CS (arrow).

**Figure 3 medicina-59-01146-f003:**
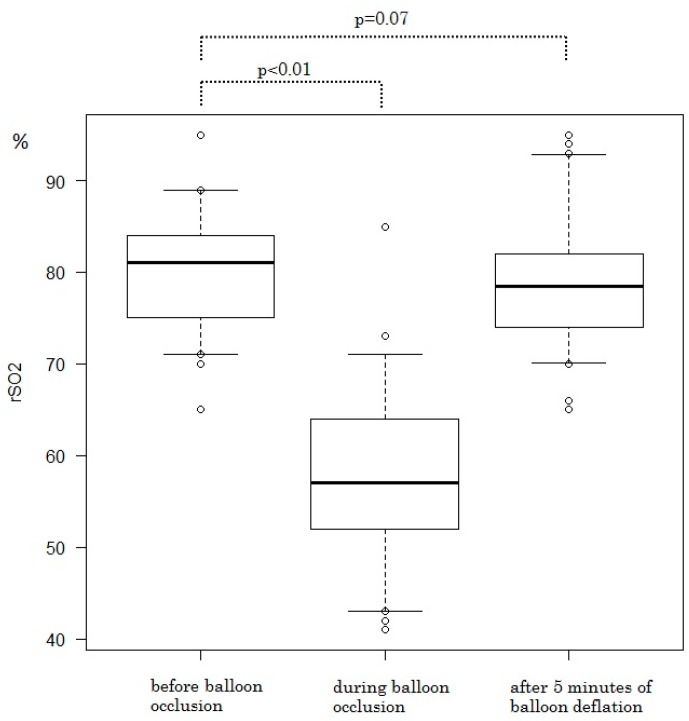
The rSO2 during balloon occlusion was significantly lower than rSO2 before balloon occlusion (57.9% ± 9.6% vs. 80.3% ± 6.0%; *p* < 0.01). The rSO2 increased in all cases after balloon deflation. Additionally, there were no significant differences between rSO2 before balloon occlusion and rSO2 after 5 min of balloon deflation (80.3% ± 6.0% vs. 78.7% ± 6.6%; *p* = 0.07). rSO2, regional oxygen saturation.

**Table 1 medicina-59-01146-t001:** Patient backgrounds and maternal outcomes.

*N*	15
age (years)	34.6 ± 5.1
parity (n)	1.6 ± 0.9
previous cesarean section (n)	1.3 ± 0.6
degree of placental adhesion (n)	
accreta	9
increta	4
percreta	2
operation time (min)	233.4 ± 111.3
estimated blood loss (mL)	3179.2 ± 244.6
hysterectomy (n)	9
number of packed RBC transfusions (units)	7.2 ± 4.2
postoperative hospital stay (days)	9.1 ± 2.9

**Table 2 medicina-59-01146-t002:** Laboratory results of the serum tests of the patients.

	Before Operation	After Operation	*p*-Value
serum creatinine (mg/dL)	0.51 ± 0.2	0.51 ± 0.1	0.50
serum potassium (mmol/L)	3.4 ± 0.4	3.8 ± 0.4	0.069
serum lactate levels (mmol/L)	1.2 ± 1.1	1.3 ± 1.0	0.64

## Data Availability

The data presented in this study are available on request from the corresponding author.

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
