# Peer review of "rSO2 Measurement Using NIRS for Lower-Limb Blood Flow Monitoring and Estimation of Safe Balloon Occlusion/Deflation Time in Patients with PAS Who Underwent PBOA during CS"

_medicina, 2023, doi:10.3390/medicina59061146_

Round 1

Reviewer 1 Report

This is an excellent on monitoring lower limb circulation in women undergoing procedure for PAS. This gives a great potential for monitoring extremities circulation in long lasting OB procedures. . As I understand, authors are not OB/GYN  specialist therefore specific comments on PAS treatment options are not in order. However, I would like to see in discussion some comments on other possibilities of use. How did You estimate acute blood loss?  NIRS potential of rSO2 evalutaion of lower limbs during PBOA for PAS has excellent potential of determination of leg ischemia. I find this article excellent.

Author Response

Reviewer 1

This is an excellent on monitoring lower limb circulation in women undergoing procedure for PAS. This gives a great potential for monitoring extremities circulation in long lasting OB procedures. . As I understand, authors are not OB/GYN specialist therefore specific comments on PAS treatment options are not in order. However, I would like to see in discussion some comments on other possibilities of use. How did you estimate acute blood loss?  NIRS potential of rSO2 evalutaion of lower limbs during PBOA for PAS has excellent potential of determination of leg ischemia. I find this article excellent.

Thank you very much for your careful review and kind comments.

The following sentences were added in discussion.

In the field of obstetrics and gynecology, NIRS may be useful for monitoring lower extremity perfusion when postpartum hemorrhage is managed with PBOA.

The following sentences were added in Materials and Methods (Data analysis).

The EBL was measured based on the volume of the suction canisters in the operating room and the weight of the surgical pads.

Reviewer 2 Report

‘’ rSO2 Measurement Using NIRS for Lower-Limb Blood Flow Monitoring and Estimation of Safe Balloon Occlusion/Deflation Time in Patients with PAS who Underwent PBOA During CS’’

The manuscript presented for review consists of 8 pages with 13 references. 2 tables and 3 figures are included. The figures and tables are appropriate. They properly show obtained data. The study is original. The manuscript is divided into 4 sections (Introduction, Material and Methods, Results, Discussion). The work fits the journal scope. The manuscript is clear, well-structured and relevant for the field. English language is fine. References are recent. The aim of the study is defined. Keywords are adequate and refer to the whole context. The results of the study and conclusions are consistent. Inclusion and exclusion criteria were described. In the section – Introduction- Authors explained a significance of conducting a study („PAS disorders has increased (…) with the increasing numer of cesarean sections”). They also highlited complications which may be connected with PAS („severe hemorrhage, hysterectomies , blood transfusion”). The presented work is of great scientific value.  

Author Response

Reviewer 2

The manuscript presented for review consists of 8 pages with 13 references. 2 tables and 3 figures are included. The figures and tables are appropriate. They properly show obtained data. The study is original. The manuscript is divided into 4 sections (Introduction, Material and Methods, Results, Discussion). The work fits the journal scope. The manuscript is clear, well-structured and relevant for the field. English language is fine. References are recent. The aim of the study is defined. Keywords are adequate and refer to the whole context. The results of the study and conclusions are consistent. Inclusion and exclusion criteria were described. In the section – Introduction- Authors explained a significance of conducting a study („PAS disorders has increased (…) with the increasing numer of cesarean sections”). They also highlited complications which may be connected with PAS („severe hemorrhage, hysterectomies , blood transfusion”). The presented work is of great scientific value.

Thank you very much for your careful review and kind comments.
